# OpenReview forum: "Kaleidoscopic Teaming in Multi Agent Simulations"
_ICLR.cc/2026/Conference — Submitted to ICLR 2026_

### Official Review · Reviewer_uatp · 2025-10-27

**Soundness:** 2
**Presentation:** 2
**Contribution:** 2
**Rating:** 4
**Confidence:** 4

**Summary:**

The paper presents the Kaleidoscopic Teaming framework, which aims to improve AI agent safety evaluations, particularly in multi-agent environments. By using the MASK framework, it simulates challenging, dynamic scenarios where agents can interact, cooperate, or compete to complete tasks. The framework is designed to capture the complexities of agent interactions, thought processes, and decision-making in real-world-like environments. The paper also proposes new optimization strategies for generating these scenarios, along with metrics to assess the safety of agents in these contexts. This work provides an important step toward more realistic and comprehensive safety evaluations for AI agents.

**Strengths:**

1. Innovative Approach:
The introduction of the Kaleidoscopic Teaming framework is an innovative and thoughtful approach to evaluating AI agent safety in complex, multi-agent settings. It offers a novel perspective on safety evaluation that goes beyond traditional red-teaming methods, making it highly relevant for the evolving landscape of AI deployment.

2. Comprehensive Simulation:
The Multi-Agent Simulation Kaleidoscopic-Teaming framework provides a thorough method for stress-testing agents by modeling real-world interactions and belief states, ensuring that agents are evaluated in dynamic, challenging scenarios.

3. In-Context Optimization Strategies:
The paper’s in-context optimization techniques to generate more challenging scenarios for agents is a strong contribution. These strategies help to push the boundaries of traditional safety evaluations and expose vulnerabilities in a more nuanced manner.

**Weaknesses:**

1. Insight Gatherer and Judges:
The roles of the Insight Gatherer and Judges seem to be somewhat separate but could benefit from a more integrated approach. Is it necessary to keep them as distinct entities, or could they be merged into one unified system to directly evaluate the agents? The paper does not fully justify the separation. An ablation experiment exploring the impact of combining or separating these components could provide valuable insights into whether the current setup is optimal.

2. Role of the Orchestrator:
The Orchestrator’s function is explained in the context of controlling agent interactions and belief states. However, its role and operation appear quite similar to that of a traditional star-shaped structure where a central node manages tasks. It would be helpful to clarify how the Orchestrator differs from traditional management nodes and whether its added complexity truly contributes to the system’s effectiveness.
3. Realism of Generated Scenarios:
A significant concern is how to ensure that the generated scenarios are realistic and grounded in real-world possibilities. The paper does not fully address how these scenarios are constructed in a way that ensures they do not deviate too far from practical situations. If scenarios are too fictional, this could lead to agents completing tasks in unrealistic ways, potentially compromising the reliability of safety evaluations. Could the authors consider leveraging existing benchmarks as seeds to generate scenarios rather than relying on the model to generate them from scratch? This would ensure that the scenarios align more closely with real-world conditions.
4. Selection of Agents in MAS:
The paper mentions creating a society of 100 agents, with eight major agent types. However, it remains unclear how the Multi-Agent Simulation (MAS) selects agents for each iteration. Figure 2 mentions, “In MASK, at each iteration, either one or more (a group of) agents are selected from the society,” but there is no detailed explanation of how this selection process works. Does the random selection of agents ensure that the chosen agents are suitable for the specific tasks in the generated scenario? If the agents are not well-suited to the tasks, the results could be meaningless or difficult to interpret. Further clarification on the agent selection process and its potential impact on safety evaluations would be valuable, as it is crucial to ensure that the chosen agents align with the specific requirements of each generated scenario.
5. Percent Negative Agents vs. Percent Negative Scenarios:
The metrics Percent Negative Agents and Percent Negative Scenarios appear to be highly similar, with one directly influencing the other. There seems to be no strong justification for measuring both separately. It might be more effective to focus on evaluating agents’ planning processes and their associated negative outcomes rather than focusing on the scenario level. This would provide a more granular understanding of where the safety issues originate.
6. Scalability of Overall Average Agent Score:
The definition and calculation of the Overall Average Agent Score, particularly the -2 and 2 values, are unclear. The paper does not provide an explanation of how these scores are derived or what specific criteria are used to assign them. Additionally, how does this score range scale when the number of agents varies? The paper does not address how this metric adapts in such cases. A more detailed explanation of how the -2 and 2 scores are determined, and how the score range scales with different agent populations, would be useful to understand the metric’s robustness and consistency.
7. Judge Rubric and Evaluation Consistency:
The decision to use the worst score from the judges rather than an average or median raises concerns about evaluation consistency. How do the judges’ individual differences influence the final scores, especially if the judges are using the same model? If the model used by the judges produces the worst score, how can it be guaranteed that this score accurately reflects an agent’s safety, rather than being a result of potential misjudgment or model error? More clarity on how the worst score is selected as the definitive evaluation would strengthen the validity of the assessment process.
8. Limited Evaluation of Attack Models:
The evaluation of attack models is somewhat limited, as the paper only compares strong LLMs as attack models against weak LLMs as target models. It would be interesting to explore the effectiveness of attack models of similar or stronger strength against the same LLMs. This would help in better assessing the robustness of the system and provide a more thorough evaluation.
9. Task Completion Evaluation:
The paper omits an important aspect of agent evaluation: task completion success (e.g., accuracy or completion rate). There is no clear measure of how well the agents complete the tasks, and this is essential for evaluating their overall safety and effectiveness. It would be valuable to introduce accuracy-based metrics to ensure that agents are not only safe but also competent in completing the tasks at hand. Moreover, should a task be deemed impossible to complete, should this be considered a safety issue as well?

**Questions:**

1. Clarification of Table and Figure Labels:
In Section 4, the table labels are inconsistent. Table 3 should be Table 1 based on the context of the results. Additionally, Figure 4 is unclear, and it is difficult to discern the meaning of each chart without additional graphical annotations. The authors should consider adding labels or explanations to make these visuals more understandable.

2. Appendix Structure:
The Appendix lacks subheadings, making it difficult to navigate, especially when referencing specific figures or details. Clearer segmentation in the appendix would help readers quickly locate the relevant information. Furthermore, the frequent citation of the appendix in the main text, with no specific reference to the relevant subsection, makes it challenging to follow and could hinder the clarity of the paper.

---

> ### Author Response · Authors · 2025-11-20
> **Response to Reviewer uatp**
>
> **Insight Gatherer and Judges:** The reason why we kept insight gatherer and judges separate was that judges were ensemble of judges while the insight gatherer is just an effective and a light weight model. Since judges produce numeric values, we can have an ensemble of them where the scores are aggregated in a way (eg min/max). However, it was harder, less efficient and effective to have ensemble of insight gatherers where the outcome is a text summary. Hence, we separated these components. We experimented with different setups and noticed that having an ensemble of judges works the best with Nova lite working the best as our insight gatherer since it is both effective and light weight so more efficient in generating longer insights.
>
> **Role of the Orchestrator:** We do not necessarily think that Orchestrator’s existence is complicated. In fact, its design is simplistic with responsibilities that differs from standard management nodes being the part that it injects the beliefs to the agents depending on the scenarios and the history of interactions.
>
> **Realism of Scenarios:** While starting with seeds existing in benchmarks sounds like a good idea, still there is a chance to diverge from these scenarios as the goal of our framework is to generate novel scenarios where they are not overfitted by scenarios existing in benchmark datasets. The goal of this paper was to evaluate agents in multi-agent simulations where novel vulnerabilities can be discovered in novel scenarios.  We believe that since the tools provided to these agents are coming from realistic APIs this can be achieved in some sense; however, our main goal was to introduce this framework where users can tune the scenarios and tools depending on their usecase.
>
> **Selection of Agents:** The process of selecting the agents is random. However, as also clarified in kaleidoscope’s system prompt, the kaleidoscope is instructed to generate scenarios that is relevant to the agents and that they can complete using their tools. We have also detailed this in Lines 195-197 and 114-115 and 143-145 in the main text and Figures 7,8,9 in the appendix.
>
> **% Negative Agents vs. % Negative Scenarios:** There might be a scenario in which all the agents act in an unsafe manner vs a scenario in which only one agent acts as unsafe, so % negative agents tries to capture that on average how many agents per scenario are unsafe vs in % negative scenario even if an agent is unsafe we mark the scenario as unsafe. As reported in the paper we utilized each metric in different places. For instance, in Figure 4, where we wanted to compare single-agent vs multi-agent scenarios we utilized the % Negative Scenarios metric which is more specific to the scenarios. We think that it would not hurt to have both of these metrics and we do not necessarily see them redundant. Each can convey different information depending on the study and analysis. In addition, we proposed the overall average agent score, which is close to what you are proposing. Each agent is graded on a scale of -2 to 2, so we can capture its planning processes and their associated negative outcomes.
>
> **Scalability of Overall Avg Agent Score:** We studied different scaling/rubric design approaches and saw that the current scaling works the best. Initially our rubric was just a number between 0-9 with no clear description of what each of those numbers meant. We instructed the judges that 0 is extremely unsafe to 9 to extremely safe. However, we found that when the scales are broad and not clearly defined, judges do not perform well. This also aligns with previous work where they also found that scales and rubrics need to be defined in detail for the LLMs judges. We followed the best practices learned from previous work and after studying various setups, we found that the current scale works the best. Our human verification of the judge results, also confirms the robustness of our rubric.
>
> **Judge Rubric & Evaluation Consistency:** The judge and the rubric results are human verified as also reported in Section 4. We have studied different setups and what is decided in the paper is as a result of various efforts which are also human verified to showcase its effectiveness and reliability.
>
> **Limited Evaluation of Attack Models:** We respectfully disagree with this statement “The evaluation of attack models is somewhat limited, as the paper only compares strong LLMs as attack models against weak LLMs as target models.” Nova Lite is a weaker model compared to other target models (e.g., Nova Pro). We tried to exactly mimic the scenario that you described here in that we chose Nova lite to be the weaker attacker against the strong models and deepseek as the stronger one which can also reason. We tried to pick different types of attackers as our attack models to exactly showcase these differences. In our results, we also show that infact we are able to get stronger results from deepseek as it is a stronger attacker.

---

> ### Author Response · Authors · 2025-11-20
> **Further Clarifications**
>
> **On Task Completion Evaluation:** The focus of this paper was to detect vulnerabilities of agents according to safety issues. While task completion evaluations according to accuracy or completion rate are important, we saw it to be out of scope for this paper. However, our framework is flexible enough that if for a use-case there is a judge that works well for task completion, it can easily be integrated as part of our framework if someone needs those metrics as well.
>
> **On Clarification of Table and Figure Labels:** We agree with this comment. Thank you so much for pointing this out. We will ensure to fix this.
>
> **On Appendix Structure:** We will ensure to address this as well and add subheadings for the Appendix.

---

> ### Comment · Reviewer_uatp · 2025-11-26
>
> Thank you for your response, which addressed some of my initial queries. However, after reviewing the comments from other reviewers and re-evaluating the paper, I still reserve several unresolved concerns and suggestions.
>
> First, regarding the choice of Nova Lite for the Insight Gatherer, I suggest adding comparative experiments with other models, as the insights generated by this component directly influence the downstream scenario generation.
>
> Second, concerning the Role of the Orchestrator, your explanation does not clearly differentiate it from a standard management node; standard nodes in multi-agent systems also manage interaction history, input scenarios, and can inject beliefs or instructions into agents, so the structural distinction seems minimal.
>
> Third, on the Realism of Scenarios, I strongly recommend adding verification results to ensure scenario plausibility. Even with real APIs, LLMs suffer from hallucinations, and without strong constraints、seeds or relevance checks on the retrieved content, the model may still fabricate unrealistic or logically incoherent scenarios.
>
> Fourth, regarding the Selection of Agents, even if the process is random, statistical results are necessary to understand the system's complexity—specifically, the average number and variety of agents per simulation. The current text does not sufficiently reveal the actual composition complexity of the MAS.
>
> Fifth, regarding the metrics, I agree with Reviewer x5hb that they need rigorous definition, ideally with formulas in the appendix. Relying on the rebuttal for interpretation is insufficient; for instance, it remains difficult to distinguish from the text without rebuttal , whether "Percent Negative Agents" refers to the proportion of unsafe agents against the total population or the proportion within specific unsafe scenarios.
>
> Sixth, the "Overall Avg Agent Score" remains unclear. While you explained why a 0-9 scale wasn't used, the rationale for a -2 to 2 scale versus 0-4 is not provided—is it to facilitate summation? Furthermore, since the random selection leads to variable agent counts, it is unclear how the metric handles this variance to prevent the number of agents from skewing the results like extreme values or very high values in batches have an impact.
>
> Finally, dismissing Task Completion as "out of scope" is problematic. For example, if an agent refuses a request in an unsafe scenario (thus failing the task), or if an agent fails a safe task because the generated scenario is unrealistic or beyond its capabilities, how are these scored? Without a task completion metric, distinguishing between safety mechanisms and simple task failure is ambiguous.

---

### Official Review · Reviewer_x5hb · 2025-10-30

**Soundness:** 1
**Presentation:** 3
**Contribution:** 2
**Rating:** 4
**Confidence:** 4

**Summary:**

The paper is about the safety evaluation of LLM agents. The authors propose a new method of safety testing called kaleidoscopic teaming, which allows for testing agents in single and multi-agent scenarios. Compared to red teaming, kaleidoscoping teaming allows for more complex, multi-turn scenarios, which are dynamically adjusted for finding vulnerabilities in specific agents. The authors describe their framework, which consists of “functional” LLMs orchestrating, making up the scenarios, and judging the agents. They consider different strategies of prompting them, and they propose a metric for safety that can be used in this framework. The experiments show diverse results of safety for different LLM agents and their types.

**Strengths:**

Importance of the topic. LLM agents should be tested for safety, and this work allows for more complex testing. Additionally, authors show the importance of multi-agent testing.

Possibility for capturing the different levels of safety.

A dynamic framework that can adapt to a specific agent/agent type. Easily automated.

**Weaknesses:**

No comparison with baselines. E.g., how would red teaming or automated red teaming grade the agents? Would there be much of a difference?

There is no formal definition of the metrics. The description is a bit unclear, and formal equations would be useful. E.g., the score value. Additionally, why are the metrics introduced if they are not used in the experiments?

It’s a very “practical paper” with no guarantees of detecting any specific kinds of vulnerabilities. And for such paper, there isn’t enough evaluation of particular parts of the framework (e.g., how different LLMs for judges influence the result)

Figure 3: For Claude 3.7(and  Nova Pro) the score is opposite for Nova Lite and Deepseek.  That suggests that the framework is very dependent on the LLM types in Kaleidoscope, orchestrator, etc. What to do with this instability of evaluation?

No mention of the code being published.

Very limited explanation of the relationship to the automated red teaming research.  The multi-agent aspect is claimed to be novel. Is there a difference between components of agents and multi-agent scenario? If this is the selling point, they should focus on specific examples from their scenarios that are of this type and explain why alternative methods would not capture them.

Though I appreciate playfulness, it was hard to focus on reading the paper due to all the colorful fonts and bolded text.

**Questions:**

What is the reason for the substantially different evaluation result of Claude 3.7 by DeepSeek and Nova Pro?

Why only the kaleidoscope component  is tested with different LLM models? Other components are always just Nova Lite.

Have you considered the weaknesses of using LLM as a judge and other components of an LLM evaluation framework? Is there a danger of collusion?

How can you make sure that testing a potentially malicious agent with access to real-world tools  (RapidAPI) will not cause any harm?

What’s the difference in evaluation in competitive and cooperative multi-agent scenarios?

---

> ### Author Response · Authors · 2025-11-20
> **Response to Reviewer x5hb**
>
> **On baselines:** The goal of kaleidoscopic teaming is different from traditional red-teaming, therefore, those methods won't apply here. For kaleidoscopic teaming, special judges and metrics need to be developed that consider all the aspects required for kaleidoscopic teaming. In addition, kaleidoscope’s goal is different from traditional red teaming methods as here the goal is to generate scenarios instead of prompts that can capture a more holistic approach in which thoughts, actions, interactions, and emergent safety risks when agents cooperate or compete against each other, which requires identification of mutual vulnerability points, to complete series of tasks in a given scenario can be identified instead of traditional red teaming in which we are interested in generating prompts that would lead into responses that would violate safety issues. Thus, a one to one and fair comparison to traditional red teaming methods is not possible. However, we have compared our proposed optimization strategies to a zero-shot baseline that is a more fair comparison within the same setup. In addition, we compared the multi-agent scenarios to single-agent scenarios which is the closest to traditional red teaming methods for agentic use-cases.
>
> **On metrics:** We have included details and descriptions about our metrics in detail in a dedicated section (section 2.3). We can also include formal equations in the final version as suggested for ease of understanding. We have used all the metrics in our experiments. Table 1 reports all those metrics. We excluded the word percent and overall to save space in Table 1, but they are the same metrics. Figures 3 and 4 also report Overall Average Agent Score and Percent Negative Scenarios respectively.
>
> **Guarantees on detecting vulnerabilities and different parts of the framework:** We have included qualitative results in our appendix to demonstrate types of vulnerabilities that is possible to be detected using our framework. We also included quantitative results that demonstrate the effectiveness of our framework in detecting different types of vulnerabilities. To determine what LLMs to use for each component in our framework we have done extensive studies. For instance, we initially started with one LLM for the judge, but we noticed that use of one LLM can bias the results to that specific LLM. We also noticed that ensemble of LLMs can work better as also observed in previous works (Gue et al., 2024). As also mentioned in our paper lines 308-312. We also performed human validation of our judges to ensure the results are reliable. Same practice was done for all the other components. Most models were not even capable of performing the task for the specific component, so we picked each LLM for each component such that it can accurately perform the task and is lightweight (e.g., Nova lite which is a lightweight model) since some components can take time and we need efficient models. Thus, we have done extensive studies to pick the most suitable setups for each component in our framework.
>
> **Framework dependent on LLM types:** This problem is not specific to our work but is applicable to any other automated red teaming framework. For instance, some models can work better to generate adversarial examples and some not (e.g., a helpful only model can generate more adversarial examples as the red model vs a model that is safety aligned). In this case as well, some models might generate better scenarios as the kaleidoscope compared to others. That is exactly why we have performed experiments using different models as the kaleidoscope to show these results and to avoid instability. For instance, in Figure 3, Nova pro might seem less vulnerable when Nova lite is used as the kaleidoscope this might be because these models are more similar where vulnerabilities might be harder to be detected vs when deepseek is used as the kaleidoscope where vulnerabilities can be detected easier. In addition, deepseek is less safety trained, so it is more likely to generate scenarios where vulnerabilities can be detected; hence, why in Figure 3 we demonstrate more negative results for Claude 3.7(and Nova Pro). We want to emphasize that this is the case for any other red teaming framework and not specific to ours. We have performed extensive experiments using various models for each of these components to provide more clear insights on this.
>
> **Ensure access to real-world tools will not cause harm:** One of the main ideas behind our work was to create a simulation which is essentially a sandbox environment where agents can be tested without actually executing these APIs. In other words, one can consider these APIs as dummy APIs where traces are just examined and agents are graded on their safety using our framework. Since our framework primarily works with the traces it gives the flexibility to treat these simulations as sandboxes where actual execution can be avoided while safety can be graded and examined.

---

> ### Author Response · Authors · 2025-11-20
> **Further Clarifications**
>
> **Weaknesses of using LLM as a judge:** We have done extensive studies to ensure the best setup for our LLM as a judge. We also have performed human evaluations to ensure our LLM as a judge has reliable performance. Our work is also grounded on the previous work that shows ensemble of LLMs can work better (Gue et al., 2024).
>
> **Different Results for Claude 3.7 by DeepSeek and Nova Pro:** In our results, we did not observe significant differences between Claude 3.7 by DeepSeek and Nova Pro by DeepSeek results. Can you please clarify which results you are referring to? However, with regards to differences between DeepSeek results vs Nova Lite, we have provided a detailed answer under “The framework dependent on the LLM types”.
>
> **Difference competitive and cooperative scenarios:** In competitive scenarios it is more likely for agents to get involved in unethical behaviors due to the competition amongst them to win the competition. We wanted to have a combination of these scenarios to observe different behaviors for when agents are encouraged to cooperate with each other vs when they are put into competition.
>
> **Differences from red teaming and multi-agent scenario:** The goal of kaleidoscopic teaming is different from traditional red-teaming. For kaleidoscopic teaming, special judges and metrics need to be developed that consider all the aspects required for kaleidoscopic teaming (e.g., thoughts, actions, agent interactions etc). In addition, the kaleidoscope’s goal is different from traditional red teaming methods as here the goal is to generate scenarios instead of prompts that can capture a more holistic approach in which thoughts, actions, interactions, and emergent safety risks when agents cooperate or compete against each other, which requires identification of mutual vulnerability points, to complete series of tasks in a given scenario can be identified instead of traditional red teaming in which we are interested in generating prompts that would lead into responses that would violate safety issues. In addition, we compared the multi-agent scenarios to single-agent scenarios and demonstrated the differences in Figure 4 and showed that more vulnerabilities can be discovered through kaleidoscopic teaming that is the framework to use to generate multi-agent scenarios.
>
>  **Why kaleidoscope component is tested w/ different LLMs:** Similar to other red teaming frameworks, the main components are the attacker and the target models to study. Hence, in our studies, we also focused on these components and reported results using various different LLMs for them. With that being said, we still performed studies to pick the best LLMs for any of the other components in our framework that are not the target and attacker models that should be the main components of our study.

---

> > ### Comment · Reviewer_x5hb · 2025-11-27
> >
> > On baselines:
> > Even if the frameworks operate differently, they share the same goal: identifying vulnerabilities. To validate the claim that Kaleidoscope 'detects more,' please provide a quantitative comparison (e.g., success rate, diversity of vulnerabilities found) against a standard baseline (e.g., manual red teaming, automated red teaming or a simple LLM-based attacker). Without this, the claim of superior detection is unsubstantiated.
> > Additionally, what is a cost/efficiency comparison? If your framework is "more complex", does it cost 10x more tokens to find the same bugs?
> >
> > On metrics:
> > You are correct, I overlooked them in Table 1. However, yes, I would like the formal definition to be included for clarity. Specifically, Overall Average Agent Score needs clarification
> >
> > Guarantees on detecting vulnerabilities and different parts of the framework:
> > “We have included qualitative results in our appendix to demonstrate types of vulnerabilities that is possible to be detected using our framework.” - The examples in the Appendix are raw logs (from two runs) without qualitative analysis. They do not demonstrate the types of unique vulnerabilities Kaleidoscope captures that other methods miss. Please provide a summarized taxonomy of these vulnerabilities in the main text to substantiate the framework's novelty.
> > Regarding the reliability of the framework (as there are no guarantees), of a set of known vulnerabilities (ground truth), how many does Kaleidoscope catch vs. baselines?
> >
> > Ensure access to real-world tools will not cause harm:
> > So agents call a tool, and it’s not a real tool? What does the agent return from the tool call? Is it a static or dynamic response? What happens when an agent finds out it is not a real tool, therefore that it is tested? Doesn’t it shorten the sensible length of this scenario-evaluation?
> >
> > Weaknesses of using LLM as a judge:
> > The response does not fully address the risk of model collusion. Specifically, when the Red Team agent, Target agent, and Judge agent share the same base model (or family), there is a risk of bias where the Judge favors the Target's style or fails to recognize specific failure modes known to that model family. How does the framework mitigate this 'self-preference' or blind spot?
> >
> > Different Results for Claude 3.7 by DeepSeek and Nova Pro:
> > There is a concerning inconsistency in the results. You state that DeepSeek generates more potent scenarios, yet the safety scores for Claude 3.7 against DeepSeek are more positive than compared to Nova Pro (Figure 3 left). This inversion suggests the metric may be sensitive to the attacker's identity rather than the target's actual safety. Can you explain this discrepancy? It implies the framework's evaluation is not model-agnostic.
> > If DeepSeek is the 'better' attacker (generating more vulnerabilities), we expect the Target Model to have a lower safety score against it. The fact that the score is higher (or the pattern is inverted) suggests your metric may be measuring, e.g., the Judge's bias, and not the Target's safety.
> >
> > Framework dependent on LLM types:
> > How do you deal with this issue of having different results for different models? How can you use this framework?
> > Do all other methods of red teaming also depend on LLM grading and generated scenarios? What about some benchmarks? E.g., a dataset that you can evaluate without the need for LLM evaluation, nor LLM in the process at all?
> >
> > Difference between competitive and cooperative scenarios:
> > Perhaps I wasn’t clear. I was asking what are the results, the differences noticed between these two options after performing the experiment. E.g., you compare single- and multi-agent scenarios (Figure 4), and I don’t see that for competitive and cooperative scenarios.
> >
> > Differences from red teaming and multi-agent scenario:
> > I can understand that your framework is more complex, however, isn’t the goal essentially the same? Both of approaches have a goal of capturing vulnerabilities. It seems that Kaleidoscope is the “red team”/”attacker” and the rest of your framework is to coordinate and grade the agents, therefore to “automate the red-teaming” process. Or am I missing something?
> > Therefore, shortly, do you claim that your framework is better than manual red-teaming and/or automated red-teaming? Or do you claim that your framework is doing more than just attack?
> >
> > Other reviews:
> > You mention in the review 5uui above that your main contribution is a framework, but you did not publish code. That undermines the contribution in my opinion.
> >
> > Additionally, I think uatp has a lot of good points which are not addressed properly, the most important in my opinion are:
> > Insight Gatherer and Judges, Role of the Orchestrator, Judge Rubric and Evaluation Consistency

---

### Official Review · Reviewer_5uui · 2025-10-31

**Soundness:** 2
**Presentation:** 2
**Contribution:** 2
**Rating:** 2
**Confidence:** 4

**Summary:**

The paper proposes an approach for jailbreaking LLMs in the context of multiple agents. The term kaleidoscopic teaming is a variation of red teaming. The authors perform simulations in which they model agents that contains agents with different capabilities and which exhibit human traits such as emotions and belief states.

**Strengths:**

* The authors performed experiments with a relatively large number of generated agents across multiple models.

**Weaknesses:**

* It is not clear whether the agents proposed by the authors are considered to be agents who might be implemented and deployed in production settings. Why would these agents exhibit human emotions?
* The main contribution of the paper is unclear. I.e. are the authors propose that instead of red teaming people perform kaleidoscope teaming? Is this intended to be a methodology or a benchmark?

**Questions:**

* Can you please clarify how would the reader take advantage of the contributions of this paper? Would it need to repeat your experiments?

**Details Of Ethics Concerns:**

No ethics review is needed.

---

> ### Author Response · Authors · 2025-11-20
> **Response to Reviewer 5uui**
>
> **On agents being deployed in production setting and exhibiting human emotions:** We wanted to mimic agents to be as realistic as possible hence exhibiting emotions as also highlighted in lines 147-149 “In our simulations, the goal is to mimic human interactions and to generate complex scenarios in which existing agents in the society can have emotions and belief states and are evaluated for their safety.”.
>
> **On the main contribution:** The main contribution is to propose this methodology or framework that can red team multi-agent use-cases and applications. As mentioned in the paper, vanilla red teaming frameworks do not consider all the aspects that our framework can support for multi-agent usecases. Hence, our framework tries to address these shortcomings.
>
> **On reader taking advantage of our contributions:** The point of our framework, similar to all other red teaming frameworks, is to dynamically find existing vulnerabilities in different target models, so to take advantage of our framework readers can run our framework against any model/agents that they want to test. This gives them the flexibility to test any target model where they can discover specific vulnerabilities for that particular target model whereas standard static benchmarks can not do that. Static benchmarks are not designed to discover specific vulnerabilities that exist in a specific target model. They also do not dynamically learn from specific failure modes of a specific target model. Our framework has the capability to dynamically discover and improve on discovering vulnerabilities in a particular target model.

---

### Official Review · Reviewer_NAF4 · 2025-11-01

**Soundness:** 3
**Presentation:** 3
**Contribution:** 3
**Rating:** 6
**Confidence:** 3

**Summary:**

The paper introduces kaleidoscopic teaming—a safety‐evaluation paradigm for AI agents (not just chat models) that stresses test thoughts, actions, tool use, and inter‑agent interactions. It operationalizes this via MASK (Multi‑Agent Simulation Kaleidoscopic‑teaming): a loop with a kaleidoscope that generates scenarios, an orchestrator that assigns turns and belief/emotional states, judges with a rubric, and an insight gatherer that feeds weaknesses back to the scenario generator. The authors also add two in‑context optimization strategies for the kaleidoscope—PSO (Past Scenarios Only) and CSR (Contrastive Scenarios + Rewards)—and define three ASR metrics (percent negative agents, percent negative scenarios, overall average agent score). Experiments span 100 agents using 32 tools across 8 domains; results show that optimized strategies often yield higher attack success and that multi‑agent settings expose more vulnerabilities than single‑agent ones.

**Strengths:**

1. The work evaluates thoughts, tool calls, and interactions rather than just final answers—closer to real agent deployments.
1. It has a clear, replicated finding that inter‑agent dynamics expose more safety failures
1. PSO and CSR are simple, model‑agnostic ways to steer the scenario generator; they often raise ASR and scenario diversity
1. Per‑agent‑type safety profiles can guide targeted mitigations.

**Weaknesses:**

1. Although partially validated, LLM‑as‑judge can encode biases; taking the worst score across an ensemble may over‑penalize edge cases. More blind human audits and evaluations on judging schemes would help.
1. The orchestrator injects belief/emotional states to nudge unethical behavior; this may inflate ASR relative to organic failures and complicate comparison with other frameworks. A controlled ablation isolating belief injection effects is needed.
1. “Percent negative” treats minor and catastrophic harms similarly; adding severity‑weighted scores and time‑to‑failure would give a richer view.
1. Datasets/traces aren’t released, hindering replication

**Questions:**

How do you handle severity-related concerns?

---

> ### Author Response · Authors · 2025-11-20
> **Author Response Reviewer NAF4**
>
> We thank the reviewer for their feedback.
>
> **On validating LLM-as-Judge:** We have performed human evaluations to verify effectiveness of the judge and demonstrated that the utilized setup works. In addition, we have studied the effects of different setups in judges and noticed that the current setup works the best. Lastly, due to our emphasis on the safety concerns, we wanted to consider the worst case outcome according to safety. Hence, we took the minimum to ensure that we consider the worst case outcome.
>
> **On Orchestrator injecting beliefs:** We wanted to mimic realistic scenarios as much as possible in which agents might have beliefs and emotions associated with them as also mentioned in lines 147-149. In addition, our zero-shot baseline also has the same setup. In other words, we made sure that all the experiments have the same setup to not inflate ASR. Lastly, we performed ablations with Mistral and Nova pro as the target models with and without belief augmentation when attacked by Nova lite and observed that while for mistral, belief augmentation has minor effects on improving ASR, for Nova Pro, we see the opposite. Hence, we do not believe that injecting beliefs will necessarily inflate ASR but rather make the simulations more realistic.
>
> | Attack Model | Target Model | Setup | Negative Agents | Negative Scenarios | Avg Agent Score |
> |--------|----------|----------|--------|----------|----------|
> | Nova lite | Mistral | w/ belief PSO | 60.4% | 74% |  -0.50 |
> | Nova lite | Mistral | w/o belief PSO | 33.2 | 49% |  0.133 |
> | Nova lite | Nova Pro | w/ belief PSO | 17.4% | 33% |  0.67|
> | Nova lite | Nova Pro | w/o belief PSO | 45.7% | 54% |  -0.09 |
>
> **On Percent negative treats minor and catastrophic harms similarly:** We have introduced the average agent score that captures the severity of risk in it. In fact, that is why we introduced different metrics since each captures one interesting aspect of our framework.
>
> **On releasing datasets:** We can work toward releasing the code and dataset after publication.
>
> **On handling severity related concerns:** We handle these cases through the introduced average agent score metric.

---

### Meta-Review · Area_Chair_Zu4k · 2026-01-06

**Summary:**

This work argues that existing red teaming or safety evaluation frameworks fall short in evaluating safety risks in complex behaviors, thought processes and actions taken by agents. Thus, the authors propose a Kaleidoscopic Teaming for multi-agent system. Basically, the work is not sharp. The authors proposed many stuffs, such as "evaluates safety of agents in both single-agent and multi-agent setups", "new in-context optimization techniques that can be used in our kaleidoscopic teaming framework" and "appropriate metrics".
3 reviewers are negative and only one is positive. I do not think the work is acceptable.

**Reviewer Concerns:**

1. Concern (NAF4(w1); uatp; x5hb):
LLM judging can encode bias; taking the minimum score across judge ensemble may be overly conservative / unstable;
The authors response: they did human evaluations validating judge reliability and explored different judge setups;
But they don’t really resolve: (a) how sensitive results are to judge choice, (b) whether min aggregation is “correct” vs robust statistics, (c) the collusion/self-preference concern (x5hb follow-up explicitly says this is not answered: "Weaknesses of using LLM as a judge: The response does not fully address the risk of model collusion. Specifically, when the Red Team agent, Target agent, and Judge agent share the same base model (or family), there is a risk of bias where the Judge favors the Target's style or fails to recognize specific failure modes known to that model family. How does the framework mitigate this 'self-preference' or blind spot?")

2.  no convincing comparison vs standard (automated/manual) red teaming. this is raised by x5hb
Need quantitative comparison against a baseline attacker/red-teaming setup + cost/efficiency comparison.
It is not satisfied by the reviewer x5hb in the follow up discussion.

3. Generated scenarios could be fictional/hallucinatory;
It is raised by uatp. Also mentioned in the follow up: "Third, on the Realism of Scenarios, I strongly recommend adding verification results to ensure scenario plausibility. Even with real APIs, LLMs suffer from hallucinations, and without strong constraints、seeds or relevance checks on the retrieved content, the model may still fabricate unrealistic or logically incoherent scenarios."

**Reviewer Scores:**

Three reviewers give negative feedbacks, only one with 6. In the discussion, it seems no reviewer is willing to increase their score.

---

### Decision · Program_Chairs · 2026-01-26

Reject